# Association of maternal depression and anxiety with toddler social-emotional and cognitive development in South Africa: a prospective cohort study

Lauren C Shuffrey ,[1,2] Ayesha Sania,[1,2] Natalie H Brito,[3] Mandy Potter,[4] Priscilla Springer,[5] Maristella Lucchini ,[1,6] Yael K Rayport,[6,7] Carlie Du Plessis,[8] Hein J Odendaal,[9] William P Fifer[1,6]

LCS and AS are joint first authors.

For numbered affiliations see end of article.

**Correspondence to**
Dr Lauren C Shuffrey;
lcg2129@cumc.columbia.edu

## ABSTRACT

**Objective** A robust literature has identified associations between prenatal maternal depression and adverse child social-emotional and cognitive outcomes. The majority of prior research is from high-income countries despite increased reporting of perinatal depression in low/ middle-income countries (LMICs). Additionally, despite the comorbidity between depression and anxiety, few prior studies have examined their joint impact on child neurodevelopment. The objective of the current analysis was to examine associations between prenatal maternal depression and anxiety with child social-emotional and cognitive development in a cohort from the Western Cape Province of South Africa.

**Design** Prenatal maternal depression and anxiety were measured using the Edinburgh Postnatal Depression Scale and the State-Trait Anxiety Inventory Scale at 20–24 weeks' gestation. Child neurobehaviour was assessed at age 3 using the Brief Infant-Toddler Social Emotional Assessment and the Bayley Scales of Infant Development III Screening Test (BSID-III ST). We used linear regression models to examine the independent and joint association between prenatal maternal depression, anxiety and child developmental outcomes.

**Results** Participants consisted of 600 maternal-infant dyads (274 females; gestational age at birth: 38.89 weeks±2.03). Children born to mothers with both prenatal depression and trait anxiety had higher social-emotional problems (mean difference: 4.66; 95% CI 3.43 to 5.90) compared with children born to mothers with no prenatal depression or trait anxiety, each condition alone, or compared with mothers with depression and state anxiety. Additionally, children born to mothers with prenatal maternal depression and trait anxiety had the greatest reduction in mean cognitive scores on the BSID-III ST (mean difference: −1.04; 95% CI −1.99 to −0.08).

**Conclusions** The observed association between comorbid prenatal maternal depression and chronic anxiety with subsequent child social-emotional and cognitive development underscores the need for targeting mental health support among perinatal women in LMICs to improve long-term child neurobehavioural outcomes.

### Strengths and limitations of this study

► The current study included a prospective evaluation of maternal depression and anxiety symptoms during pregnancy and prospective assessment of cognitive and social-emotional outcomes within a large cohort of South African mother–children pairs.
► Limitations include a lack of data on maternal mental health assessments postnatally and mother–child dyadic measures, which are potential mediators of the relationship between prenatal maternal depression, prenatal maternal anxiety and child developmental outcomes.
► This study addresses a significant gap in the literature in research examining the impact of prenatal maternal psychological health on child neurobehavioural development in resource-poor communities.

## INTRODUCTION

Decades of research on the early origins of behaviour has promoted the concept that the prenatal environment has a profound impact on one's risk for the development of neurodevelopmental or psychiatric disorders.[1 2] Several prior studies have identified associations between prenatal maternal depression and increased risk for social-emotional problems and decrements in cognitive development. However, the majority of prior research in this domain is from high-income countries (HICs), despite increased reporting of perinatal depression in low/middle-income countries (LMICs) including South Africa where an estimated 35% of women report prenatal depression.[3 4] Additionally, despite the comorbidity between depression and anxiety, few prior studies have examined their joint impact on child neurodevelopment.

Several studies in HICs have identified associations between prenatal maternal

depression, anxiety, and offspring behavioural, social-emotional, and cognitive development.[5] Specifically, a meta-analysis demonstrated adverse effects of prenatal maternal depression and anxiety on child social-emotional problems, with ORs of 1.79 and 1.50, respectively.[6] Prenatal maternal depression and anxiety are also associated with cognitive and language deficits,[7 8] delayed motor development,[8] emotional and behaviour dysregulation,[9–11] inattention and hyperactivity[12–14] and difficult temperament.[15] A more recent meta-analysis not only confirmed prior reports, but also found that the effects of perinatal maternal depression extend beyond infancy through adolescence.[16]

The developmental origins of health and disease (DOHaD) model posits that maternal psychological distress during pregnancy (eg, perceived stress, depression, anxiety, post-traumatic stress) may result in changes in hypothalamic pituitary adrenal (HPA) axis function and upregulation of inflammatory processes with downstream effects on offspring perinatal brain development and behaviour.[17] Prior research suggests comorbid prenatal maternal depression and anxiety may be associated with the greatest increases in maternal HPA-axis activity and differential changes in immunological activity. Specifically, comorbid prenatal maternal depression and anxiety have been associated with a greater increase in salivary cortisol levels,[18] TH1 secreted cytokines, TH2 secreted cytokines, and TH17 secreted cytokines[19] compared with either condition alone. Other research suggests cytokine profiles may differ between individuals with prenatal maternal depression and anxiety.[19–21]

Risk factors may be exacerbated in LMICs such as South Africa, where both poverty and perinatal mental health disorders are highly prevalent.[22 23] Specifically, in South Africa maternal mood disorders have been linked to structural and community stressors associated with markers of poverty including less than a high school education, lack of social support, alcohol use, family stress, food insecurity, lack of partner involvement and intimate partner violence.[24–28] There are few prior studies that have examined the impact of prenatal maternal mental health on offspring social-emotional or cognitive development in South Africa. While few prior studies reported significant harmful effects of prenatal maternal stress or perinatal maternal depression, there is still a significant gap in the literature in research examining the impact of prenatal maternal psychological health on child neurobehavioural development in resource-poor communities.[24 29 30] Additionally, to our knowledge no prior South African studies have examined the joint effect of prenatal maternal depression and anxiety on child neurobehavioural outcomes. Research examining the long-term impact of prenatal maternal depression and anxiety on child behavioural and cognitive outcomes is critical for providing justification to local public health services for targeting mental health support in perinatal women from underserved communities.

The objective of the current analysis was to determine if prenatal maternal depression and state or trait anxiety were associated with child social-emotional problems or cognitive development at approximately 3 years of age in a South African cohort from the Western Cape.

## MATERIALS AND METHODS

### Participants

Participants were a subset of infants with available outcome data at age 3 enrolled in the Safe Passage study conducted by the Prenatal Alcohol and SIDS and Stillbirth Network, a multi-centre study investigating the role of prenatal exposure in risk for sudden infant death syndrome, stillbirth and fetal alcohol spectrum disorders. Eligibility criteria for the Safe Passage study included the ability to provide informed consent in English or Afrikaans, 16 years of age or older at the time of consent, and a gestational age between 6 weeks and 40 weeks at the time of consent based on estimated delivery date.[31] Exclusion criteria for prenatal maternal enrolment into the Safe Passage study included planned therapeutic abortion, moving out of the catchment area prior to estimated date of delivery and clinical judgement.

### Maternal assessments

#### Maternal-infant chart abstraction, demographic and socioeconomic measures

Maternal-infant medical charts were abstracted to obtain maternal age at delivery, gestational age at birth, mode of delivery and the infant's biological sex. Measures to collect prenatal alcohol, tobacco and recreational drug exposure have been previously described.[31 32] Prenatal maternal alcohol and tobacco use behaviours were previously characterised using cluster analysis.[33 34] Through study specific case report forms, participants indicated demographic and socioeconomic information including race, maternal educational attainment, household crowding (persons per room in household), access to running water inside the house, prenatal care during pregnancy and marital status.

#### Self-reported depression and anxiety measures

Information regarding maternal mental health during pregnancy was obtained at 20–24 weeks' gestation. Depressive symptoms were measured using the Edinburgh Postnatal Depression Scale (EPDS), a depression screening tool developed to specifically assess depressive symptoms in perinatal women where higher scores indicate more severe depressive symptoms.[35 36] The EPDS is widely used and has been validated in English and Afrikaans in South Africa.[35 37] Prior studies have used a cut-off score of ≥12 or ≥13 to be indicative of major depression within perinatal women living in South Africa.[35 37] Maternal anxiety symptoms were measured using the State-Trait Anxiety Inventory (STAI),[38] an anxiety screening tool to distinguish anxiety symptoms from depressive symptoms which has also been validated in both languages.[39] The STAI has two subscales, state anxiety which reflects the participant's current state of anxiety when completing

the questionnaires and trait anxiety, which is thought to be consistent across time and reflect personality traits. In HICs, the STAI has a cut-off score of ≥40 on both the state anxiety and trait anxiety subscales to indicate a threshold for clinical levels of anxiety. Based on these prior studies, we used a cut-off of ≥13 to indicate maternal depression, a cut-off of >40 on the STAI-state subscale to indicate state anxiety and a cut-off of >40 on the STAI-trait subscale to indicate trait anxiety.

## Toddler developmental assessments
### Bayley Scales of Infant Development III Screening Test
The Bayley Scales of Infant Development III Screening Test (BSID-III ST) were designed as a rapid assessment of cognitive, language and motor functioning in infants and young children in order to determine if a child's development is within normal limits and identify risk for developmental delay. The BSID-III ST has high test–retest reliability: cognitive (0.85), receptive language (0.88), expressive language (0.88), fine motor (0.82) and gross motor (0.86). Although the BSID-III ST does not identify degree of impairment, the cut-off points indicate whether a child shows competence in age-appropriate tasks, evidence of emerging age-appropriate skills and evidence of being at risk for developmental delay. The BSID has been validated and widely used throughout South Africa.[40 41]

### Brief Infant-Toddler Social Emotional Assessment
The Brief Infant-Toddler Social and Emotional Assessment (BITSEA) is a 42-item parental report measure of social-emotional development, behavioural problems and delays in competence.[42] Domains assessed within the BITSEA include: externalising (activity/impulsivity, aggression/defiance, peer aggression), internalising (depression/withdrawal, anxiety, separation distress, inhibition to novelty), dysregulation (sleep, negative emotionality, eating, sensory sensitivity) and competence (compliance, attention, imitation/play, mastery motivation, empathy and pro-social peer relations).[42] Findings from the BITSEA validation study provide preliminary support for the BITSEA as a reliable and valid brief screener for infant-toddler social-emotional and behavioural problems in addition to delays in competence.[43] When used in socioeconomically and ethnically diverse community-based populations, the BITSEA demonstrated excellent test–retest reliability and good inter-rater agreement between parents.[42]

## Statistical analyses
Using multiple linear regression models, we estimated independent and joint effects of maternal depression and state and trait anxiety on social-emotional problem, social emotional competence and cognitive development scores. Two, separate four-level categorical prenatal maternal mental health variables were created to assess the impact of prenatal maternal depression, trait anxiety and state anxiety. We created a prenatal maternal

depression and trait anxiety variable with four categories: (1) no prenatal depression or trait anxiety (n=199; 33.17%, reference category), (2) prenatal depression only (106; 17.67%), (3) prenatal trait anxiety only (n=68; 11.33%) and (4) prenatal maternal depression and trait anxiety (n=227; 37.83%) (table 1). In separate models we additionally examined the independent and joint effects of prenatal maternal depression and state anxiety. We created a prenatal maternal depression and state anxiety variable with four categories: (1) no prenatal depression or state anxiety (n=248; 41.33%; reference category), (2) prenatal depression only (n=237; 39.50%), (3) prenatal state anxiety only (n=19; 3.17%) and (4) prenatal maternal depression and state anxiety (n=96; 16%) (table 1). For each regression model, either no prenatal maternal depression or trait anxiety or no prenatal maternal depression and state anxiety was set as the reference category. Minimally adjusted models included sex, gestational age at birth and age at follow-up as covariates. Fully adjusted models additionally controlled for prenatal maternal alcohol use, prenatal maternal tobacco use, maternal employment status at delivery, maternal educational attainment at delivery, parity and the household crowding index. We used missing indicator methods and median imputation to account for missing categorical and continuous covariate data, respectively (described in table 1). All analyses were performed in SAS software V.9.4 (SAS Institute).

## RESULTS
### Maternal and child demographic characteristics
The participants included in the present analysis consisted of mothers and their infant born between April 2014 and August 2015 from the Western Cape Province of South Africa who participated in a follow-up study to examine social-emotional development and cognitive development at approximately 3 years of age. A total of n=18 mother–infant dyads were excluded due to missing maternal prenatal mental health data. The final sample consisted of 600 maternal-infant dyads (274 females; gestational age at birth: 38.89 weeks±2.03) (table 1).

### Child social-emotional development
Based on the BITSEA problem scale percentile rank score of 26 or higher, 51% of children (306/600) were classified as having a 'possible problem'. Based on the BITSEA competence scale percentile rank score of 15 of lower, 5% (30/600) of children were classified in the 'possible deficit/delay range' for social competencies (table 2). There were no significant sex differences in social-emotional problems on the BITSEA, however, girls had significantly higher social-emotional competence compared with boys (mean difference: 0.38, 95% CI 0.05 to 0.71, p=0.03).

**Table 1** Sociodemographic characteristics

| | Mean±SD or N (%) | N and percent missing |
|---|---|---|
| **Maternal characteristics** | | |
| Maternal age (years) | 25.26±5.91 | 0 (0%) |
| Maternal body mass index (BMI) | | 0 (0%) |
| BMI<18.5 (kg/m$^2$) | 62 (10.33%) | |
| BMI 18.5–25 (kg/m$^2$) | 282 (47%) | |
| BMI 25–30 (kg/m$^2$) | 132 (22%) | |
| BMI>30 (kg/m$^2$) | 124 (20.67%) | |
| Parity | | 0 (0%) |
| Parity<1 | 218 (36.33%) | |
| Parity=1 | 196 (32.67%) | |
| Parity=2 | 103 (17.17%) | |
| Parity≥3 | 83 (13.83%) | |
| Antenatal care visits | | 0 (0%) |
| Antenatal care visit <3 | 49 (8.17%) | |
| Antenatal care visit 3–6 | 389 (64.83%) | |
| Antenatal care visit >6 | 162 (27%) | |
| Caesarean section | 106 (17.70%) | 1 (0.1%) |
| Education | | 0 (0%) |
| Some primary school | 43 (7.167%) | |
| Some high school | 405 (67.57%) | |
| Completed high school | 119 (19.83%) | |
| Beyond high school | 33 (5.5%) | |
| Married | 290 (48.33%) | 0 (0%) |
| Employed | 172 (28.67%) | 0 (0%) |
| Adjusted household crowding | 1.56±0.72 | 2 (0.33%) |
| Depression (Edinburgh≥13) | 333 (55%) | 0 (0%) |
| Anxiety (State-Trait Anxiety Inventory≥40) | 115 (19.17%) | 0 (0%) |
| Maternal prenatal alcohol use cluster groups | | 0 (0%) |
| Non-drinking group: 0 standard drinks/trimester | 245 (46.93%) | |
| Moderate-high continuous drinking group | 122 (23.37%) | |
| Standard drinks in trimester 1 | 27±39 | |
| Standard drinks in trimester 2 | 17±25 | |
| Standard drinks in trimester 3 | 9.4±15 | |
| Binge drinking events (>4 drinks/day) trimester 1 | 2.7±4 | |
| Binge drinking events (>4 drinks/day) trimester 2 | 1.7±2.8 | |
| Binge drinking events (>4 drinks/day) trimester 3 | 0.89±1.7 | |
| Low continuous drinking | 26 (4.98%) | |
| Standard drinks in trimester 1 | 1.4±2.5 | |
| Standard drinks in trimester 2 | 4±2.8 | |
| Standard drinks in trimester 3 | 0.62±1.1 | |

Continued

**Table 1** Continued

| | Mean±SD or N (%) | N and percent missing |
|---|---|---|
| Binge drinking events (>4 drinks/day) trimester 1 | 0.067±0.25 | |
| Binge drinking events (>4 drinks/day) trimester 2 | 0.30±0.46 | |
| Binge drinking events (>4 drinks/day) trimester 3 | 0±0 | |
| Quit early drinking | 129 (24.71%) | |
| Standard drinks in trimester 1 | 8.5±6.5 | |
| Standard drinks in trimester 2 | 0.31±0.87 | |
| Standard drinks in trimester 3 | 0.056±0.31 | |
| Binge drinking events (>4 drinks/day) trimester 1 | 0.84±0.82 | |
| Binge drinking events (>4 drinks/day) trimester 2 | 0±0 | |
| Binge drinking events (>4 drinks/day) trimester 3 | 0±0 | |
| Maternal prenatal tobacco use | | 2 (0.33%) |
| Non-smoking (0 cigarettes/trimester) | 227 (37.97%) | |
| Moderate-high continuous smoking | 132 (22.07%) | |
| Average cigarettes in trimester 1 | 45±20 | |
| Average cigarettes in trimester 2 | 50±27 | |
| Average cigarettes in trimester 3 | 48±25 | |
| Low continuous smoking | 222 (37.12%) | |
| Average cigarettes in trimester 1 | 16±9.4 | |
| Average cigarettes in trimester 2 | 16±9.7 | |
| Average cigarettes in trimester 3 | 16±10 | |
| Quit early smoking | 17 (2.84%) | |
| Average cigarettes in trimester 1 | 11±7.5 | |
| Average cigarettes in trimester 2 | 0.15±0.32 | |
| Average cigarettes in trimester 3 | 0.079±0.11 | |
| Raw Maternal Edinburgh Score | 12.99±5.73 | 0 (0%) |
| Raw Maternal State Anxiety Score | 31.23±10.24 | 0 (0%) |
| Raw Maternal Trait Anxiety Score | 40.63±10.63 | 0 (0%) |
| Depression–trait anxiety groups | | 0 (0%) |
| No depression or trait anxiety | 199 (33.17%) | |
| Depression alone | 106 (17.67%) | |
| Trait anxiety alone | 68 (11.33%) | |
| Depression and trait anxiety | 227 (37.83%) | |

Continued

**Table 1** Continued

| | Mean±SD or N (%) | N and percent missing |
|---|---|---|
| Depression–state anxiety groups | | 0 (0%) |
| No depression or state anxiety | 248 (41.33%) | |
| Depression alone | 237 (39.50%) | |
| State anxiety alone | 19 (3.17%) | |
| Depression and state anxiety | 96 (16%) | |
| HIV status | | |
| Tested for HIV | 600 (100%) | |
| HIV positive | 1 (0.1%) | |
| **Infant characteristics** | | |
| Infant sex | | 0 (0%) |
| Male | 326 (54.33%) | |
| Female | 274 (45.67%) | |
| Gestational age at birth (weeks) | 38.89±2.03 | 0 (0%) |
| Infant birth weight (g) | 2980.65±564.65 | 2 (0.33%) |
| Follow-up age (months) | 38.29±2.96 | 0 (0%) |
| Adjusted follow-up age (months) | 38.14±0.016 | 0 (0%) |

**Table 2** Neurodevelopmental outcome raw scores and at-risk groups

| | |
|---|---|
| BSID-III Screening Test language of administration | |
| English only | 177 (30%) |
| Afrikaans only | 360 (62%) |
| Mixed English and Afrikaans | 48 (8%) |
| BSID-III Screening Test scores | |
| Gross motor | 25.47±4.54 |
| Fine motor | 23.55±4.08 |
| Cognitive | 27.77±4.99 |
| Problem solving | 13.40±6.77 |
| Receptive language | 21.69±5.10 |
| Expressive language | 21.27±3.78 |
| At-risk categories | |
| Gross motor | 21 (4%) |
| Fine motor | 18 (3%) |
| Cognitive | 24 (4%) |
| Receptive language | 33 (6%) |
| Expressive language | 25 (4%) |
| Brief Infant-Toddler Social-Emotional Assessment (BITSEA) | |
| Social emotional problem | 13.40±6.77 |
| Competence | 19.56±2.07 |
| At-risk categories | |
| Social emotional problems | 306 (51%) |
| Social-emotional competence | 30 (5%) |

BSID-III, Bayley Scales of Infant Development III.

## Association between prenatal maternal depression, trait anxiety and child social-emotional development

Compared with children born to mothers with no prenatal depression or trait anxiety, children born to mothers with prenatal depression and trait anxiety had higher social-emotional problems (mean difference: 4.66; 95% CI 3.43 to 5.90), followed by prenatal maternal trait anxiety only (mean difference: 3.87; 95% CI 2.07 to 5.66) and finally prenatal maternal depression only (mean difference: 2.76; 95% CI 1.23 to 4.29) in minimally adjusted models. These associations remained significant in fully adjusted models where similarly comorbid prenatal maternal depression and trait anxiety group was associated with the highest child social-emotional problems (mean difference: 4.33; 95% CI 2.90 to 5.67), followed by prenatal maternal trait anxiety only (mean difference: 3.23; 95% CI 1.19 to 5.27), with the smallest mean difference for the prenatal maternal depression only group (mean difference: 2.64; 95% CI 1.02 to 4.27) as compared with the no prenatal depression or trait anxiety group (figure 1). Additional significant predictors in the multivariate models were parity of 3 or greater which was associated with lower social-emotional problems (mean difference: −3.02, 95% CI −4.63 to −1.40) and low continuous smoking during pregnancy which was associated with higher social-emotional problems (mean difference: 1.39, 95% CI 0.039 to 2.74). There were no significant associations between prenatal maternal depression, trait anxiety and child social-emotional competence.

## Association between prenatal maternal depression, state anxiety and child social-emotional development

Compared with children born to mothers with no prenatal depression or state anxiety, children born to mothers with comorbid prenatal depression and state anxiety had higher social-emotional problems (mean difference: 4.29; 95% CI 2.73 to 5.84). Children born to mothers with prenatal depression only also had higher social-emotional problems compared with mothers with no prenatal depression or state anxiety (mean difference: 2.71; 95% CI 1.51 to 3.88). These associations remained significant in fully adjusted models (prenatal depression and state anxiety: 3.90 mean increase (95% CI 2.19 to 5.60); prenatal depression only: 2.58 mean increase (95% CI 1.34 to 3.82)) (figure 1). Additional significant predictors in the multivariate models were parity of 3 or greater which was associated with lower social-emotional problems (mean difference: −3.22, 95% CI −4.86 to −1.58), low continuous smoking during pregnancy which was associated with higher social-emotional problems (mean difference: 1.44; 95% CI 0.075 to 2.81) and finally less than a high school education (some primary school only) which was associated with higher social-emotional problems

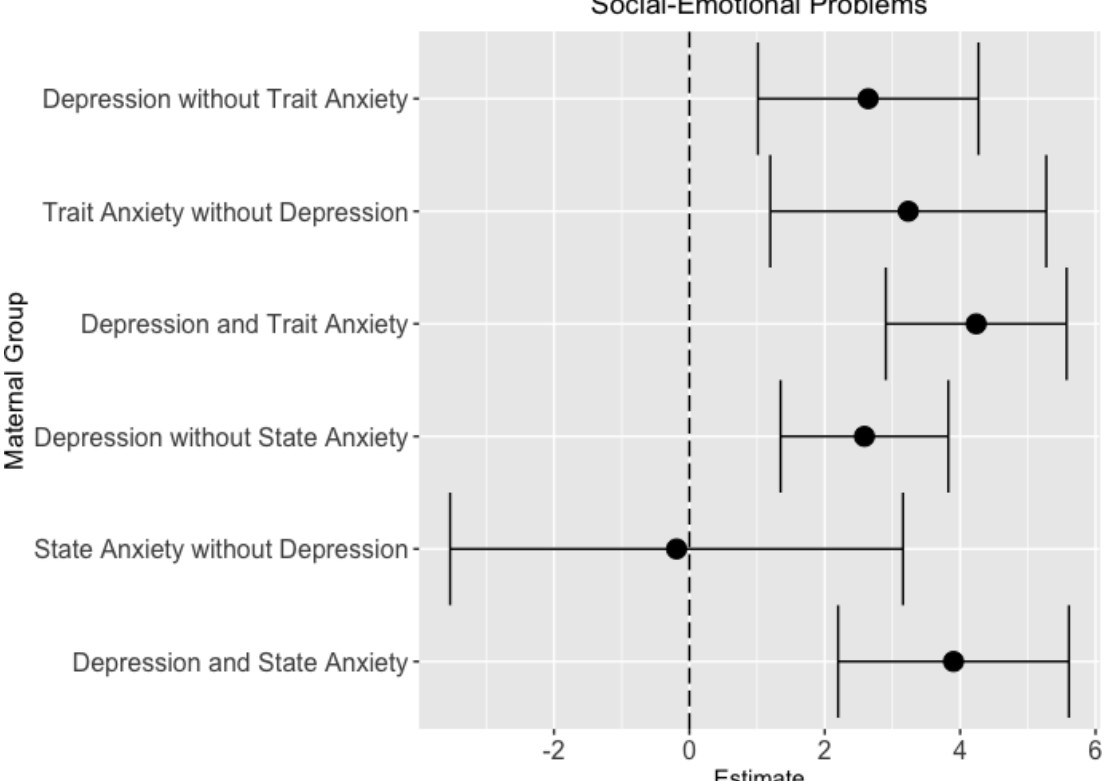

**Figure 1** Association between prenatal maternal depression, prenatal maternal anxiety and child social-emotional problems measured by the Brief Infant-Toddler Social and Emotional Assessment. Each line plot depicts the mean difference and their CI for child social-emotional problems (x-axis) for each prenatal maternal mental health group (y-axis). Either no prenatal maternal depression and state anxiety or no prenatal maternal depression and trait anxiety were the reference groups. Models were adjusted for sex, gestational age at birth, age at follow-up, prenatal maternal alcohol use, prenatal maternal tobacco use, maternal employment status at delivery, maternal educational attainment at delivery, parity and the household crowding index.

(mean difference: 3.47; 95% CI 0.248 to 6.70). However, there was no significant association between prenatal state anxiety only and child social-emotional problems on the BITSEA. There were also no significant associations between prenatal maternal depression, state anxiety and social-emotional competence.

### Child cognitive, language and motor development
Based on normative cognitive cut-off scores defined by the BSID-III ST, 4% of children (24/600) were classified as at-risk (table 2). Risk classification percentages were similar across all subdomains, expressive language: 4%; receptive language: 6%; gross motor: 4% and fine motor: 3%. There were no significant sex differences in child cognitive scores.

### Association between prenatal maternal depression, trait anxiety, and child cognitive, language, and motor development
Compared to children born to mothers with no prenatal depression or trait anxiety, children born to mothers with comorbid prenatal depression and trait anxiety had lower cognitive scores on the BSID-III ST (mean difference: −1.04; 95% CI −1.99 to −0.08). Results remained significant in the fully adjusted model (mean difference: −1.11; 95% CI −2.13 to −0.09) (figure 2) and in posthoc analyses where we additionally controlled for language

of administration for the BSID-III ST (mean difference: −0.51, 95% CI −0.99 to −0.042). Children who were assessed on the BSID-III ST in Afrikaans (mean difference: −1.00, 95% CI −1.48 to −0.52) or who were assessed in mixed English and Afrikaans (mean difference: −1.30, 95% CI −2.07 to −0.54) has significantly lower cognitive scores compared with children assessed in English. Low continuous prenatal maternal alcohol use was also associated with lower cognitive scores (mean difference: −1.30; 95% CI −2.36 to −0.24).

### Association between prenatal maternal depression, state anxiety, and child cognitive, language, and motor development
Compared with children born to mothers with neither prenatal depression nor trait anxiety, there was no significant association between cognitive scores for children born to mothers with prenatal depression only, prenatal trait anxiety only, or combined prenatal depression and trait anxiety in either the minimally or fully adjusted models.

### DISCUSSION
In summary, we found the greatest increase in child social-emotional problems in children born to women

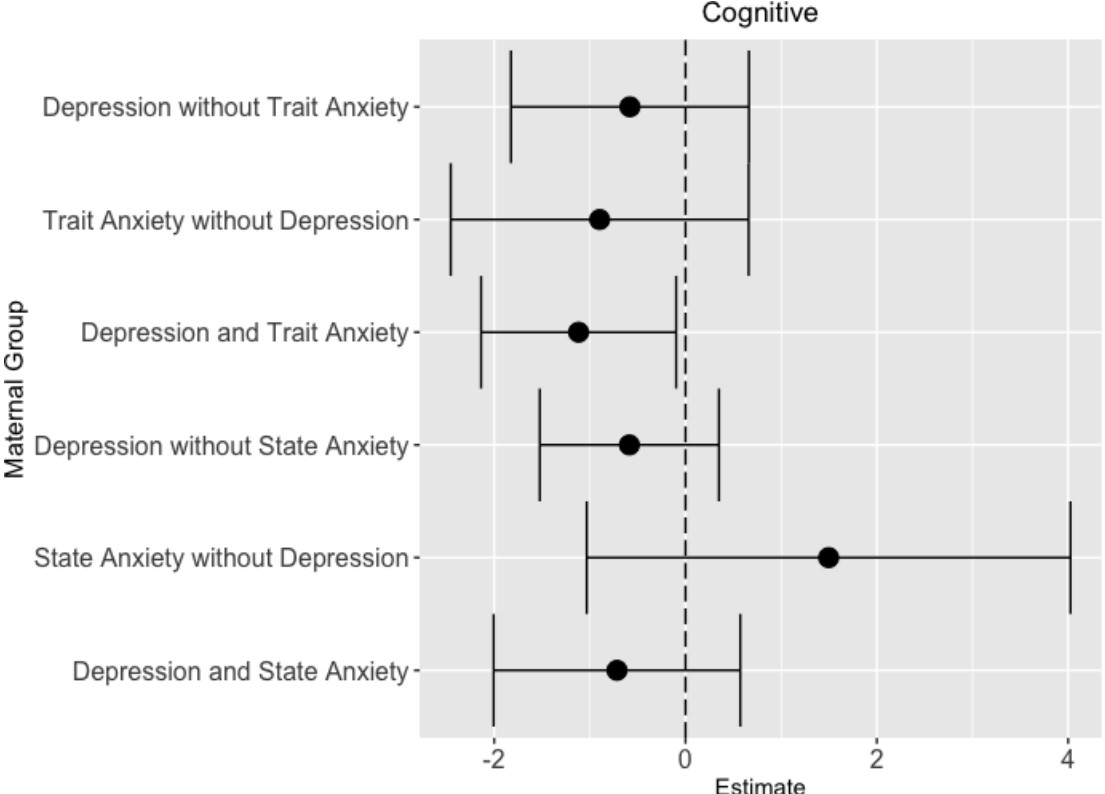

**Figure 2** Association between prenatal maternal depression, prenatal maternal anxiety and child cognitive development measured by the Bayley Scales of Infant Development III Screening Test. Each line plot depicts the mean difference and their CI for child cognitive development (x-axis) for each prenatal maternal mental health group (y-axis). Either no prenatal maternal depression and state anxiety or no prenatal maternal depression and trait anxiety were the reference groups. Models were adjusted for sex, gestational age at birth, age at follow-up, prenatal maternal alcohol use, prenatal maternal tobacco use, maternal employment status at delivery, maternal educational attainment at delivery, parity and the household crowding index.

with comorbid prenatal depression and trait anxiety compared with women with no prenatal depression or trait anxiety, prenatal depression alone or prenatal trait anxiety alone. We additionally found a significant association between comorbid prenatal maternal depression and state anxiety on higher child social-emotional problems; however, we found no association between prenatal maternal state anxiety in the absence of prenatal maternal depression on child social-emotional problems. Finally, we reported children born to mothers with prenatal maternal depression and trait anxiety had lower cognitive scores on the BSID-III ST, but we did not find an association between prenatal maternal depression alone on cognitive outcomes at 3 years of age.

Our finding linking prenatal maternal depression and anxiety to social-emotional risk parallels two recent South African studies linking increased prenatal maternal stressors to higher behavioural problems in children with an OR of 2.52, but not before 4 years of age[29] and increased perinatal depression to aggressive behaviours at 60 months of age.[24] Our findings are also largely consistent with a recent meta-analysis of studies predominately conducted in HICs, which found prenatal maternal depression and anxiety were associated with higher social-emotional problems in offspring with larger effect sizes for prenatal maternal depression (OR 1.79; 95% CI 1.61 to 1.99)

compared with prenatal maternal anxiety (OR 1.50, 95% CI 1.36 to 1.64).[6] We found the greatest increase in child social-emotional problems in children born to women with comorbid prenatal depression and trait anxiety and no effect of prenatal maternal state anxiety in the absence of prenatal maternal depression on child social-emotional problems. Our results are suggestive that chronic anxiety measured via trait anxiety may be more predictive of child social-emotional outcomes than a single measurement of concurrent anxiety during pregnancy.

A prior meta-analysis[6] also found stronger effects of prenatal maternal depression and anxiety on child social-emotional outcomes when sociodemographic risk factors such as low-income, lower levels of parental education or single family households, were highest.[6] Similarly, we found that lower levels of maternal education, low continuous tobacco use during pregnancy and low continuous alcohol use during pregnancy were associated with higher social-emotional problems in children at 3 years of age. Intriguingly, maternal parity of 3 or greater was protective and associated with lower social-emotional problems which may be due to reduced depression and anxiety levels in women with higher parity, changes in the perception of their child's behaviour due to having multiple children or additional social opportunities during sibling interactions.[44 45]

The Drakenstein Child Health Study based in the Western Cape found approximately 50% of the overall sample (369/731) were categorised as having cognitive delay at 2 years of age based on cut-offs defined by US normative data.[30] Better cognitive outcomes were associated with higher maternal education, older child age, a primigravid mother and higher socioeconomic status whereas prenatal maternal depression was associated with a 1.03 SD (95% CI −1.94 to −0.12) reduction in cognitive scores on the Bayley Scales of Infant and Toddler Development at 2 years of age.[30] Similar to the Drakenstein Child Health Study,[30] we found children born to mothers with prenatal maternal depression and trait anxiety had lower cognitive scores on the BSID-III ST compared with the no prenatal maternal depression or trait anxiety group. In contrast, we did not find an association between prenatal maternal depression alone on cognitive outcomes at 3 years of age. However, since we did not use the full BSID-III and only administered the BSID-III ST, it is difficult to directly compare our results to The Drakenstein Child Health study findings. Moreover, both studies relied on US normative data to define cut-off scores.

More recently, a large home-visiting intervention study based in Cape Town examined the effect of prenatal maternal depression only, postnatal maternal depression only (birth—60 months) or recurrent prenatal and postnatal maternal depression. This study also accounted for several other risk factors including intimate partner violence, HIV status and alcohol use on child social behaviours, language skills and cognitive development. No associations were found between maternal depression at any time point with children's language or cognitive development at 36 or 60 months of age.[24] However, children of never depressed mothers had lower aggressive behaviours on the Child Behavior Checklist at 60 months of age than children of mothers with postnatal depression only or perinatal depression.[24]

There are several biological mechanisms which can explain prior studies and our current findings linking prenatal maternal depression and anxiety with child social-emotional behaviours and cognitive development such as increased prenatal maternal inflammation, increased cortisol production and/or epigenetic changes.[46–55] Pregnancy is associated with changes in inflammatory processes and increased placental cortisol production with reduced maternal HPA axis sensitivity to stress.[46 47] However, prenatal maternal depression may upregulate inflammatory processes and/or cortisol production. Additionally, prior research found women with comorbid depression and anxiety have the greatest increases in salivary cortisol levels.[18] Maternal cortisol can cross the placenta resulting in increased inflammation[46 47] and/or affect the developmental of limbic regions, which are associated with social and emotional processes.[56] In animal models, increased proinflammatory proteins have been associated with widespread changes in perinatal brain development such as with volume reductions in grey and white matter, decreased density of GABAergic neurons, reduced synaptic pruning, and network dysfunction with potential downstream effects on neurobehavioural development.[48–55] Other studies examining the intergenerational transmission of trauma have demonstrated transgenerational epigenetic changes in animal models.[57] It is also possible comorbid maternal depression and trait anxiety may indicate a specific phenotype that is genetically transmitted to the next generation resulting in psychosocial sequelae in offspring. Taken together, prior research suggests multiple overlapping pathways by which prenatal maternal mood can affect offspring brain-behavioural development.

While we report an association of prenatal maternal depression and anxiety with child social-emotional and cognitive development from a prospective cohort study with a fairly large sample of participants, there are several methodological and contextual limitations within the current study that are worth noting. First, it is important to note that the reliance on maternal-report measures to characterise child social-emotional development is a limitation in the majority of research to date, including the present study. The reliance on maternal reporting of child social-emotional development may be influenced by factors such as maternal mood or education. An additional limitation was the use of the BSID-III screening test in the current study to measure cognitive development, which is based on US normative data. Future studies should consider objective measures of child social-emotional development through observational or behavioural coding paradigms in addition to using objective cognitive developmental assessments with normative data in South African children.

There are also several unmeasured contextual factors which could affect our findings. For example, although prenatal maternal and child nutrition are known to affect child neurobehavioural development, we lacked measures of prenatal maternal nutrition, prenatal maternal micronutrient deficiencies, prenatal and postnatal household food insecurity, and information regarding child nutrition. Additionally, prior studies have emphasised the importance of measuring childhood trauma, maternal stress and social support in communities with several heterogeneous risk factors,[3] and these measures were not collected as part of the original NIH Safe Passage study or our recent follow-up study. Therefore, we could not examine their effects or consider maternal social support as a potential moderator of resilient neurodevelopmental outcomes. There is also robust literature examining both postpartum depression (PPD) and the early mother–infant relation in shaping child outcomes. Two prior studies in South Africa demonstrated maternal intrusiveness and coerciveness mediated the association between maternal PPD and early childhood attachment.[58 59] In the current study, we did not collect data postnatally between birth and 3 years of age. Additionally, we did not collect postnatal information on maternal depression, anxiety or stress. Therefore, due to this methodological limitation

we cannot draw conclusions regarding the combined effect of the prenatal and postnatal environment on child neurodevelopmental outcomes, nor can we assess potential interaction effects between prenatal and postnatal maternal mood on child social-emotional and cognitive outcomes. Finally, our results may not generalise to all South African populations where HIV rates can be as high as 35% since our cohort only included one woman with HIV. Strengths of the current analyses include our large sample size, evaluation of both cognitive and social-emotional outcomes within the same cohort, and the detailed prospective collection of both depression and anxiety symptoms during pregnancy[31]

Our results suggest comorbid prenatal maternal depression and chronic anxiety have a greater impact on child social-emotional and cognitive development than either condition alone or than comorbid prenatal maternal depression and transitory anxiety during pregnancy. These findings are supported by a robust literature within the DOHaD framework linking perturbations in the gestational environment to later neurodevelopmental or psychiatric sequelae. Our results also lend support for future intervention studies aimed at perinatal mental health interventions targeting maternal depressive and anxiety symptoms to improve long-term child social-emotional and cognitive developmental outcomes in low-resource communities.

**Author affiliations**
[1]Psychiatry, Columbia University Irving Medical Center, New York City, New York, USA
[2]Division of Developmental Neuroscience, New York State Psychiatric Institute, New York City, New York, USA
[3]Department of Applied Psychology, New York University, New York City, New York, USA
[4]Obstetrics and Gynaecology, Stellenbosch University Faculty of Medicine and Health Sciences, Cape Town, South Africa
[5]Paediatrics and Child Health, Stellenbosch University, Stellenbosch, South Africa
[6]Neuroscience, New York State Psychiatric Institute, New York City, New York, USA
[7]Department of Psychiatry, Columbia University Irving Medical Center, New York City, New York, USA
[8]Department of Obstetrics and Gynaecology, Stellenbosch University Faculty of Medicine and Health Sciences, Cape Town, South Africa
[9]Obstetrics & Gynaecology, Stellenbosch University Faculty of Medicine and Health Sciences, Cape Town, South Africa

**Contributors** WPF and HJO acquired funding for and designed the work. LCS, AS, NHB, ML, YKR, PS, HJO and WPF conceptualised the work. MP, PS, CDP and HJO acquired the data. LCS, NHB and WPF led in data collection oversight and quality control. AS, LCS and YKR analysed the data. All authors contributed to the interpretation of data and drafting the work and revising it critically for important intellectual content. All authors approved the final manuscript and agree to be accountable for all aspects of the work. LCS and AS accept full responsibility for the work, had access to the data, and controlled the decision to publish.

**Funding** This research was supported by the Bill and Melinda Gates Foundation (WPF) and grants U01HD055154, U01HD045935, U01HD055155, and U01HD045991 issued by the Eunice Kennedy Shriver National Institute of Child Health and Human Development (NICHD) and grant U01AA016501 issued by the National Institute on Alcohol Abuse and Alcoholism (NIAAA). LCS is supported by K99HD103910 issued by the Eunice Kennedy Shriver National Institute Of Child Health & Human Development. AS is supported by UH30D023279-05S1, re-entry supplement from Office of the Director, NIH, and Office of Research on Women

Health (ORWH). The content is solely the responsibility of the authors and does not necessarily represent the official views of the National Institutes of Health.

**Competing interests** None declared.

**Patient and public involvement** Patients and/or the public were not involved in the design, or conduct, or reporting, or dissemination plans of this research.

**Patient consent for publication** Not applicable.

**Ethics approval** This study involves human participants and was approved by Stellenbosch University (#N16-08-101 and N06-10-210), New York State Psychiatric Institute (#5338). Informed consent was obtained for the Safe Passage study and from a parent or legal guardian of each participant for developmental follow-up assessments.

**Provenance and peer review** Not commissioned; externally peer reviewed.

**Data availability statement** Data are available upon reasonable request.

**ORCID iDs**
Lauren C Shuffrey http://orcid.org/0000-0003-3631-5971
Maristella Lucchini http://orcid.org/0000-0002-7968-7196

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
