## [Reviewer comments · BMJ Open]

ARTICLE DETAILS

TITLE (PROVISIONAL)	Association of Maternal Depression and Anxiety with Toddler Social-Emotional and Cognitive Development in South Africa: A prospective cohort study
AUTHORS	Shuffrey, Lauren; Sania, Ayesha; Brito, Natalie; Potter, Mandy; Springer, Priscilla; Lucchini, Maristella; Rayport, Yael; Du Plessis, Carlie; Odendaal, H. J.; Fifer, William

VERSION 1 – REVIEW

REVIEWER	Donald, Kirsten recommended University of Cape Town, Division of Developmental Paediatrics, Department of Paediatrics and Child Health
REVIEW RETURNED	24-Nov-2021

GENERAL COMMENTS	This is an important report, exploring the relationship between symptomatic current as well as reported chronic anxiety and depression in mid-pregnancy with child socio-emotional and early developmental risk status at age 3 years in a South African birth cohort. This is a well-written manuscript which represents novel work in the LMIC-context and the study has a number of key strengths including prospective data collection in pregnancy for risk (including nuanced substance use documentation) and the fact that the contextual factors were adequately included in the final model, such that the reader can be reassured that the findings are less likely to be driven by other contextual risk factors. Unmeasured background factors such as intimate partner violence and postnatal mental health problems are mentioned in the limitations. The potentially important but unexplored risk factors which are not mentioned, and which are highly relevant in this context, include biological/physical factors such as maternal nutrition, micronutrient deficiencies such as Iron, maternal HIV status. Is this data available for inclusion in the model? At the very least if available, this should be included in the demographic table for illustrative context. Some specific points that require clarification. 1. The substance use categories used, are not defined in the manuscript. This would be fine as a footnote to the table. This is important as the category of “low continuous” smoking and drinking in mothers during pregnancy seems to be particularly relevant categories.2. Throughout the manuscript the authors refer to “increase” in socio-emotional or cognitive problems. This particular term suggests a longitudinal/trajectory in the outcome data which this analysis doesn’t represent. Replace with higher/lower etc throughout3. Reference to the BSID III throughout the paper isn’t technically correct. The tool used was the screener and this only denotes risk
---

	categories rather than actual developmental outcomes, so outcomes for this tool in child development need to be reframed as developmental/cognitive risk in the relevant domains. In Table 2 the developmental tool is referred to as the Mullens?? A further point regarding the use of the Bayley scales is that while it is widely used globally, comparing to US norms is not recommended. Culturally biased aspects of the tool, particularly in the language domain, issues around language of administration as well as exposure (or not) to this type of material may all contribute. Something of this should be mentioned.
--	--

REVIEWER	Karlsson, Linnea Turun Yliopisto, Institute of Clinical Medicine, FinnBrain Birth Cohort Study
REVIEW RETURNED	16-Feb-2022

GENERAL COMMENTS	The paper addresses a highly relevant topic of maternal prenatal psychological distress influencing offspring neurodevelopment in South Africa. As the authors correctly state, most of the literature in this line of research comes from Western European or North American regions or countries potentially causing bias to the knowledge we have on child development and its predictors and prerequisites. Thus, this large study population from South Africa using standardized and validated assessment instruments and study design enables important comparisons with earlier studies and adds important information on the existing data. I only have two minor suggestions:  1. I understood that concurrent and postnatal maternal symptoms of depression and anxiety were not included in the assessment? If so, the authors should describe potential sources of assessment bias in BITSEA (e.g. depressed mothers rate their children as having more problems than non-depressed). Also, other possible limitations on not knowing the continuity of maternal symptoms to the postnatal period (e.g. influence on interaction, parental stress and postnatal stress exposure effects on the child) could be discussed a bit more in detail. The authors mention the topic but more specificity regarding the findings (e.g. BSIF findings vs neurodev at 3 years) would be interesting. 2. The possibility of genetic transmission of a given phenotype should be included in the discussion: trait anxiety & depression may indicate a specific phenotype that is genetically transmitted to the next generation and appears as problems in psychosocial development or neurodevelopment. The role of actual prenatal exposure to maternal stress and inflammation is not known based on this study, albeit can be suspected. 3. I miss some more in depth reasoning and discussion on why combined depression and trait anxiety would be different from depression alone or combined with state anxiety? What was the original idea per why investigate these separately and combined? was it just that the authors expected that more is more or that there would be some specific reasons to believe that state and trait would pose different risks? Or what is known of differential influences of depression and anxiety on offspring development in general?
---

VERSION 1 – AUTHOR RESPONSE

Reviewer: 1

Prof. Kirsten Donald, University of Cape Town

Comments to the Author:

This is an important report, exploring the relationship between symptomatic current as well as reported chronic anxiety and depression in mid-pregnancy with child socio-emotional and early developmental risk status at age 3 years in a South African birth cohort. This is a well-written manuscript which represents novel work in the LMIC-context and the study has a number of key strengths including prospective data collection in pregnancy for risk (including nuanced substance use documentation) and the fact that the contextual factors were adequately included in the final model, such that the reader can be reassured that the findings are less likely to be driven by other contextual risk factors. Unmeasured background factors such as intimate partner violence and postnatal mental health problems are mentioned in the limitations.

The potentially important but unexplored risk factors which are not mentioned, and which are highly relevant in this context, include biological/physical factors such as maternal nutrition, micronutrient deficiencies such as iron, maternal HIV status. Is this data available for inclusion in the model? At the very least if available, this should be included in the demographic table for illustrative context.

- We have updated Table 1 to include that all participants were tested for HIV and N = 1 participant was HIV positive. We have added to the limitations section that our results may not generalize to all South African populations since our cohort only included N = 1 woman with HIV. Posthoc analyses excluding this participant do not alter findings.
- We unfortunately do not have biological or physical factors such as prenatal maternal or child nutrition information, prenatal maternal micronutrient deficiencies such as iron, or surveys on food insecurity. This would have been very interesting to include in our models. The lack of this information has now been added as an additional limitation in the discussion section.

Some specific points that require clarification.

1. The substance use categories used, are not defined in the manuscript. This would be fine as a footnote to the table. This is important as the category of “low continuous” smoking and drinking in mothers during pregnancy seems to be particularly relevant categories.

- We apologize for the lack of clarity. We have revised the Maternal Assessments - *Maternal-infant chart abstraction, demographic, and socioeconomic measures* to specify the following: Prenatal maternal alcohol and tobacco use behaviors were previously characterized using cluster analysis^{29 30} (Pini et al., 2019, Shuffrey et al., 2020). We have also included information regarding the mean and standard deviation of standard drinks per trimester, the mean and standard deviation of binge drinking events per trimester (≥ 4 standard drinks per day), and the mean and standard deviation of cigarettes per trimester for participants included in the present analysis for each cluster category in Table 1.

2. Throughout the manuscript the authors refer to “increase” in socio-emotional or cognitive problems. This particular term suggests a longitudinal/trajjectory in the outcome data which this analysis doesn't represent. Replace with higher/lower etc throughout

- We apologize for our lack of clarity in language. We used linear regression models, therefore we used the terms “increase” or “decrease” reflect scores for each group compared to the reference category in each analysis.
- We have replaced increased and decreased with higher and lower in the results section and throughout the majority of the discussion, except when explaining which group had the greatest increase or decrease in a particular outcome measure.

3. Reference to the BSID III throughout the paper isn't technically correct. The tool used was the screener and this only denotes risk categories rather than actual developmental outcomes, so outcomes for this tool in child development need to be reframed as developmental/cognitive risk in the relevant domains. In Table 2 the developmental tool is referred to as the Mullen's?? A further point regarding the use of the Bayley scales is that while it is widely used globally, comparing to US norms is not recommended. Culturally biased aspects of the tool, particularly in the language domain, issues around language of administration as well as exposure (or not) to this type of material may all contribute. Something of this should be mentioned.

- We apologize for our lack of clarity in language. We have replaced the BSID-III to state the BSID-III ST (screening test) throughout the manuscript.
- We have corrected our mistake in Table 2. The Mullen was not used in this study.
- We have added the following sentence to the discussion: "However, since we did not utilize the full BSID-III and only administered the BSID-III ST, it is difficult to directly compare our results to The Drakenstein Child Health."
- We also added the following sentence to the limitations: "An additional limitation was our use of the BSID-III screening test to measure cognitive development, which is based on United States normative data."
- We had not initially considered the language of BSID-III ST administration in our analyses. We thank the reviewer for this suggestion, and have now conducted additional posthoc analyses examining language of administration on the BSID-III ST. The frequency of each administration language (English vs. Afrikaans vs. mixed languages) can be found in table 1.

Reviewer: 2

Dr. Linnea Karlsson, Turun Yliopisto

Comments to the Author:

The paper addresses a highly relevant topic of maternal prenatal psychological distress influencing offspring neurodevelopment in South Africa. As the authors correctly state, most of the literature in this line of research comes from Western European or North American regions or countries potentially causing bias to the knowledge we have on child development and its predictors and prerequisites. Thus, this large study population from South Africa using standardized and validated assessment instruments and study design enables important comparisons with earlier studies and adds important information on the existing data.

I only have two minor suggestions:

1. I understood that concurrent and postnatal maternal symptoms of depression and anxiety were not included in the assessment? If so, the authors should describe potential sources of assessment bias in BITSEA (e.g. depressed mothers rate their children as having more problems than non-depressed). Also, other possible limitations on not knowing the continuity of maternal symptoms to the postnatal period (e.g. influence on interaction, parental stress and postnatal stress exposure effects on the child) could be discussed a bit more in detail. The authors mention the topic but more specificity regarding the findings (e.g. BSIF findings vs neurodev at 3 years) would be interesting.

- We have added the following paragraph to the limitations section to address this important concern: "First, it is important to note that the reliance on maternal-report measures to characterize child social-emotional development is a limitation in all prior South African literature to date, including the present study. The reliance on maternal reporting of child social-emotional development may be influenced by factors such as maternal mood or education. An additional limitation of our outcome measures was our use of the BSID-III screening test to measure cognitive development, which is based on United States normative data. Future studies should consider objective measures of child social-emotional development through observational or behavioral coding paradigms in addition to utilizing objective developmental assessments with normative data in South African children." We also added an additional statement to clarify our inability to examine interaction effects between the pre and postnatal environment and child outcomes: "Therefore, we cannot draw conclusions regarding the combined effect of the pre- and postnatal environment on child neurodevelopmental outcomes, nor can

we assess potential interaction effects between pre- and postnatal maternal mood on child social-emotional and cognitive outcomes.”

2. The possibility of genetic transmission of a given phenotype should be included in the discussion: trait anxiety & depression may indicate a specific phenotype that is genetically transmitted to the next generation and appears as problems in psychosocial development or neurodevelopment. The role of actual prenatal exposure to maternal stress and inflammation is not known based on this study, albeit can be suspected.

- We thank the reviewer for this thoughtful suggestion. We have now added the following sentences to the discussion paragraph. “Other studies examining the intergenerational transmission of trauma have demonstrated transgenerational epigenetic changes in animal models⁵⁴. Taken together, prior research suggests multiple overlapping pathways by which prenatal maternal mood can affect offspring brain-behavioral development.”

3. I miss some more in depth reasoning and discussion on why combined depression and trait anxiety would be different from depression alone or combined with state anxiety? What was the original idea per why investigate these separately and combined? was it just that the authors expected that more is more or that there would be some specific reasons to believe that state and trait would pose different risks? Or what is known of differential influences of depression and anxiety on offspring development in general?

- Data examining biological/physiological mechanisms underlying prenatal maternal depression and anxiety suggests comorbid depression and anxiety may be related to the greatest increases in HPA axis and immunologic activity. We have provided additional information to the introduction and discussion to justify why examining each condition separately, and comorbid, is important from a psychobiological perspective.

VERSION 2 – REVIEW

REVIEWER	Karlsson, Linnea Turun Yliopisto, Institute of Clinical Medicine, FinnBrain Birth Cohort Study
REVIEW RETURNED	17-Mar-2022

GENERAL COMMENTS	The authors have mainly addressed my concerns raised during the first round of comments. I suggest that the comment "the possibility of genetic transmission of a given phenotype should be included in the discussion: trait anxiety & depression may indicate a specific phenotype that is genetically transmitted to the next generation and appears as problems in psychosocial development or neurodevelopment " is still acknowledged briefly, though. the authors have added a notion on the role of epigenetic transmission, which is correct but what I meant was direct genetic transmission of phenotypes and their antecedents. I.e. a situation where prenatal stress is actually not influencing the outcome but it is direct genetic transmission that we actually see (but which is unmeasured).
--

VERSION 2 – AUTHOR RESPONSE

We apologize for our misinterpretation of the reviewer's request and thank the reviewer for their thoughtful consideration of pathways by which maternal mental health can impact the next generation.

We have now added "It is also possible comorbid maternal depression and trait anxiety may indicate a specific phenotype that is genetically transmitted to the next generation resulting in psychosocial sequelae in offspring" to the discussion section.

VERSION 3 – REVIEW

REVIEWER	Karlsson, Linnea Turun Yliopisto, Institute of Clinical Medicine, FinnBrain Birth Cohort Study
REVIEW RETURNED	18-Mar-2022
GENERAL COMMENTS	All my comments have now been adequately responded.